# Non-Cytotoxic Graphene Nanoplatelets Upregulate Cell Proliferation and Self-Renewal Genes of Mesenchymal Stem Cells

**DOI:** 10.3390/ijms25189817

**Published:** 2024-09-11

**Authors:** Natália Fontana Nicoletti, Daniel Rodrigo Marinowic, Daniele Perondi, João Ismael Budelon Gonçalves, Diego Piazza, Jaderson Costa da Costa, Asdrubal Falavigna

**Affiliations:** 1Cell Therapy Laboratory (LATEC), University of Caxias do Sul (UCS), Caxias do Sul 95070-560, Brazil; nfnicoletti@ucs.br (N.F.N.); daniel.marinowic@pucrs.br (D.R.M.); 2Brain Institute of Rio Grande do Sul (BraIns), Pontifical Catholic University of Rio Grande do Sul (PUCRS), Porto Alegre 90610-000, Brazil; joao.goncalves@edu.pucrs.br (J.I.B.G.); jcc@pucrs.br (J.C.d.C.); 3UCSGRAPHENE, University of Caxias do Sul (UCS), Caxias do Sul 95070-560, Brazil; dperondi1@ucs.br (D.P.); dpiazza1@ucs.br (D.P.); 4Health Sciences Graduate Program, University of Caxias do Sul (UCS), Caxias do Sul 95070-560, Brazil

**Keywords:** graphene, mesenchymal stem cells, cell cycle, nanomaterials

## Abstract

Graphene nanoplatelets (UGZ–1004) are emerging as a promising biomaterial in regenerative medicine. This study comprehensively evaluates UGZ–1004, focusing on its physical properties, cytotoxicity, intracellular interactions, and, notably, its effects on mesenchymal stem cells (MSCs). UGZ–1004 was characterized by lateral dimensions and layer counts consistent with ISO standards and demonstrated a high carbon purity of 0.08%. Cytotoxicity assessments revealed that UGZ–1004 is non-toxic to various cell lines, including 3T3 fibroblasts, VERO kidney epithelial cells, BV–2 microglia, and MSCs, in accordance with ISO 10993–5:2020/2023 guidelines. The study focused on MSCs and revealed that UGZ–1004 supports their gene expression alterations related to self-renewal and proliferation. MSCs exposed to UGZ–1004 maintained their characteristic surface markers. Importantly, UGZ–1004 promoted significant upregulation of genes crucial for cell cycle regulation and DNA repair, such as CDK1, CDK2, and MDM2. This gene expression profile suggests that UGZ–1004 can enhance MSC self-renewal capabilities, ensuring robust cellular function and longevity. Moreover, UGZ–1004 exposure led to the downregulation of genes associated with tumor development, including CCND1 and TFDP1, mitigating potential tumorigenic risks. These findings underscore the potential of UGZ–1004 to not only bolster MSC proliferation but also enhance their self-renewal processes, which are critical for effective regenerative therapies. The study highlights the need for continued research into the long-term impacts of graphene nanoplatelets and their application in MSC-based regenerative medicine.

## 1. Introduction

Regenerative medicine aims to replace or restore damaged, malfunctioning, or missing tissues or body systems by utilizing biological substitutes to repair and maintain physiological function. The recent surge of interest in graphene-based nanomaterials for biomedical applications has resulted in numerous promising reports in the literature over the past few years [1,2,3]. These studies suggest that graphene-based materials are highly promising for supporting the adhesion, proliferation, and differentiation of various cell types, including mesenchymal stem cells (MSCs) [4]. Since the cell cycle is crucial for cell growth, division, and viability, one of the mammalian cells’ most critical processes, understanding how graphene properties can modulate cell cycle dynamics is essential for developing advanced biomedical applications. 

Graphene is considered a disruptive technology due to its exceptional properties. Composed primarily of carbon, similar to graphite, graphene exhibits unique characteristics such as lightness, strength, and electrical conductivity. It consists of a single layer of carbon atoms arranged in a two-dimensional planar sheet with a hexagonal lattice pattern, measuring less than 100 nm in nanoscale dimensions. These remarkable properties have garnered significant attention in the biomedical sciences. Graphene-based materials can be classified based on the number and spatial arrangement of layers, oxygen content, and chemical modifications. According to ISO/TS 80004–13:2017, the terms for different graphene structures include two-layer graphene (2LG), which refers to bilayer graphene, and few-layer graphene (FLG), consisting of 3 to 10 graphene layers. Derivatives of graphene include graphene nanoplatelets (GNPs), graphene oxide (GO), and reduced graphene oxide (rGO). Graphene nanoplatelets, also known as nanoplates (NPGs), are one of the commercially available graphene derivatives. They consist of aggregated graphene sheets forming plates with a few nanometers thickness and lateral sizes ranging from 100 nm to 100 µm. GO and rGO are more commonly used in biomedical applications due to their distinct chemical properties and versatility [5]. 

The discovery of the biocompatibility of graphene and its vast potential has positioned it as an attractive biomaterial, offering new insights into tissue engineering and regenerative medicine applications. Specific graphene-based nanomaterials are compatible with human osteoblasts [6], murine fibroblasts [7], and mammalian adenocarcinoma cells [8]. Moreover, studies have demonstrated that graphene and its derivatives play a pivotal role in enhancing the proliferation and differentiation of MSCs [6], closely mimicking the cells’ physiological microenvironment properties. Cells cultured in the presence of graphene exhibit improved growth, proliferation, and differentiation, though further research is required to optimize cell viability and fully harness these effects.

Controlling cell fate has become critical in regenerative tissue engineering and medicine. Biomaterials that interact with the intracellular environment of MSCs are central to this effort. Understanding these interactions is essential for managing the factors that influence cell stimuli and behavior, ultimately translating these insights into restoring cellular and tissue functions. This research proposes the synthesis and optimization of MSC therapy enhanced by graphene nanoplatelets, positioning them as a promising candidate for development into a bioactive and compatible cell product for advanced cellular therapy. Key considerations include cell viability and proliferation profiles, cell cycle checkpoints, and DNA repair mechanisms in cells exposed to graphene. This study underscores the non-cytotoxic profile of graphene nanoplatelets and maps cell cycle regulatory genes that are crucial to cell fate, demonstrating an increase in cell proliferation and self-renewal of MSCs exposed to graphene nanoplatelets-both of which are critical for effective regenerative therapies.

## 2. Results

### 2.1. Graphene Nanoplatelets

The relative frequency distribution of the number of layers in the graphene nanoplatelet samples used in this study is shown in Figure 1C. The highest frequency is observed between two and fifteen layers, indicating that the material is classified as having multiple layers. This histogram was generated by measuring at least 200 isolated flakes in accordance with ABNT ISO/TS 21,356 standards. Thus, these results represent the general characteristics of this group of samples. The histogram of the lateral sizes of the graphene nanoplatelets shows that they range from 100 to 4000 nm. This range aligns with the definition provided in ISO/TS 80004–13:2017, which describes graphene nanoplatelets as consisting of graphene sheets that form aggregated plates with a few nanometers in thickness and lateral sizes varying from 100 nm to 100 µm. Similar to the thickness measurements, the lateral size measurements were obtained from scanning electron microscopy analyses of at least 200 flakes under current graphene characterization standards (Figure 1D). 

Figure 1A,B illustrate the characteristic morphology of the graphene nanoplatelet sample, showing multiple aggregated graphene sheets that form stacked surfaces. Figure 1E presents the Raman spectrum of the graphene nanoplatelets, displaying bands characteristic of graphite-based materials. The D Band, located near 1350 cm^−1^, is known as the disorder or defect band, which may indicate sp^3^ bonds (tetragonal configuration) or disorders in sp^2^ hybridization bonds (e.g., edge configurations in graphene). The G Band, approximately at 1580 cm^−1^, is the graphite or tangential band corresponding to sp^2^ bonds between carbon atoms in the planar configuration. The 2D Band, situated around 2700 cm^−1^, is a second-order band crucial for characterizing graphene and its derivatives [9,10]. The bands observed in the Raman spectra classify the sample as a graphite-derived material.

According to the ISO 11,308 standard, carbon purity is assessed by correlating the purity content with the residual mass left in the crucible after analysis. In this study, the residual mass of UGZ–1004 was found to be 0.08% by weight, indicating excellent carbon purity. The classification criteria for carbon purity are as follows: residual mass < 1% is considered excellent, residual mass between 1% and 5% is good, residual mass between 5% and 20% is fair, and residual mass > 20% is poor.

Figure 1F shows that the graphene nanoplatelet sample is predominantly composed of carbon (approximately 96.9% by mass) and oxygen (approximately 2.9% by mass). The high carbon content supports the high purity level observed in the carbon purity analysis. It can also be inferred that no impurities associated with inorganic elements were observed.

### 2.2. Cellular Cytotoxicity

The UGZ–1004 graphene nanoplatelets, noted for their biological functionality and environmentally friendly production, were characterized and evaluated for cytotoxicity and biocompatibility following ISO–10993–5/2020/2023 standards. The UGZ–1004 graphene nanoplatelets demonstrated no cytotoxic effects on fibroblast 3T3, epithelial kidney VERO, or microglia BV–2 cell lines, as well as on mesenchymal stem cells (MSCs), even after 72 h of direct contact. These results indicate that the material does not adversely affect the cellular environment (Figure 2).

### 2.3. Intracellular Interaction and Viability

Graphene nanoplatelets are gray, allowing their interaction with cells to be visualized without additional dyes. This visibility is enhanced by the BV–2 microglia’s monocytic origin and phagocytic activity, which facilitate interaction with exogenous particles, including graphene nanoplatelets. Light microscopy images obtained at 40× magnification (Figure 3A) illustrate this interaction. Our study shows that while microglia incorporate graphene particles, there are no significant changes in cell viability. Additionally, population doubling data indicate that the BV–2 microglia cell line can remove and clear graphene nanoplatelets from the culture environment 28 days after initial exposure (Figure 4B). It is important to note that no methods were employed to alter membrane permeability or induce this interaction. 

To investigate the intracellular interaction of graphene with microglia at the nuclear level, DAPI staining was performed on the BV–2 cell line. The analysis revealed that UGZ–1004 did not induce any morphological changes or irregularities in the nuclear structure, as assessed by morphometric analysis. The nuclei of the microglia cells appeared regular in shape with well-defined nuclear surfaces, and no alterations in proliferation or viability were observed in vitro (Figure 3B).

To further confirm the absence of cell death following seven days of UGZ–1004–microglia interaction, the LIVE/DEAD Cell Viability Assay kit was utilized (Figure 3C). Green staining indicates viable cells, while red staining indicates dead cells. The assay also allowed for the visualization of graphene nanoplatelets within the intracellular environment, demonstrating their presence without impacting cell viability (Figure 3C).

### 2.4. MSC Characteristic Preservation 

The adherence of cells to plastic and the preservation of basal proliferation rates are key indicators of cell survival. Following positive results with immortalized cell lines, the interaction of 2D graphene with MSCs was further investigated. Equine bone-marrow-derived MSCs were exposed to UGZ–1004 graphene nanoplatelets through direct contact. Initially, UGZ–1004 interacts with the cell surface via hydrogen bonds, which are crucial for the adhesion and internalization of graphene nanoplatelets. The hydrophobic nature of graphene can be modified through surface functionalization, and cellular uptake is influenced by the lateral size of the nanoplatelets, which must be small enough to be internalized without causing toxicity.

After 5 to 7 days of UGZ–1004 exposure, graphene nanoplatelets were observed in the cytoplasm of MSCs, as indicated by their grayish color visible under light microscopy (Figure 4A). Flow cytometry analysis conducted after ten days of UGZ–1004 exposure confirmed the maintenance of MSC surface markers, including anti-CD44, anti-CD90, anti-CD105, anti-CD45, and anti-CD34. These results demonstrate that mesenchymal characteristics were preserved despite exposure to graphene nanoplatelets, with no loss of MSC capabilities or differentiation potential (Figure 4A).

### 2.5. Gene Expression

After exposure to graphene, MSCs and BV–2 cells exhibited significant differences in the transcriptional behavior of certain genes (Figure 5). The five genes with the highest and lowest fold changes compared to unexposed cells were selected for presentation and discussion. In MSCs, notable upregulation was observed for genes related to self-renewal, proliferation, and survival, including CDK1 (12.07-fold), CDK2 (9.92-fold), MDM2 (6.44-fold), KPNA2 (7.42-fold), and DNA repair with MRE11A (5.6-fold) and CDC6 (7.19-fold). Downregulation was observed for the genes CCNG2 (−9.0-fold), CCND1 (−5.84-fold), TFDP1 (−5.84-fold), CDC34 (−5.14-fold), and HUS1 (−4.52-fold), which are normally linked to the development of various tumor types when overexpressed. In BV–2 cells exposed to graphene, upregulation was noted for the genes CASP3 (11.84-fold), CCNH (10.40-fold), MCM4 (10.23-fold), CCNG2 (9.46-fold), and ATM (8.78-fold). Only two genes, MDM2 (−5.69-fold) and CCNT1 (−5.91-fold), showed reduced expression.

## 3. Discussion

Despite the growing interest in nanotechnology-based materials for regenerative medicine, graphene represents a novel class of biomaterials with considerable potential for medical applications. Given its relatively recent discovery, it is imperative to thoroughly investigate the toxicological profiles of graphene nanoplatelets to ensure their safety and efficacy in future clinical applications.

Graphene-based materials, including pristine graphene sheets, few-layer graphene flakes, and graphene oxide (GO), possess unique, versatile, and tunable properties that can be exploited in biomedical fields. For instance, graphene can be functionalized with various chemical groups such as carboxyl, amine, epoxy, and hydroxyl, which act as sites for attaching specific biomolecules.

The design of graphene materials for biomedical applications necessitates a comprehensive understanding of their intrinsic properties and functionalization strategies to optimize their performance and biocompatibility. The available functionalization methods allow for modifying graphene’s properties to meet specific requirements, enhancing its applicability in diverse biomedical contexts. Noncovalent and covalent functionalization techniques are employed to improve dispersion and stability, extend circulation time, and minimize cytotoxicity. Noncovalent methods may involve the use of hydrophilic polymers or surfactants, while covalent modifications often include the attachment of carboxylic acids and other functional groups. These approaches enhance the solubility of graphene materials in biological environments and ensure their safe and effective application in medical settings [11]. Additionally, further functionalization with proteins, DNA, metal nanoparticles, and other biomolecules can augment the specificity and efficiency of graphene materials for targeted biomedical applications. These advanced modifications enable precise targeting of specific biological interactions, thereby improving the performance and efficacy of graphene-based materials in various medical applications [12].

The primary key point of this study was to evaluate the expression of genes associated with the proliferation and self-renewal of mesenchymal stem cells (MSCs) following exposure to non-toxic graphene nanoplatelets. To accomplish this, we assessed the biocompatibility of UGZ–1004, a graphene nanoplatelet biomaterial, in contact with mammalian cells, particularly MSCs. In vitro assays were employed to evaluate the cytocompatibility of UGZ–1004, focusing on parameters such as toxicity, cell viability, and proliferative capacity. Recent studies have demonstrated that the cytotoxicity of graphene and its derivatives is influenced by particle shape, size, dispersibility, and surface functionalization. Standard protocols for assessing the cytotoxic effects of carbon-based nanomaterials include the evaluation of oxidative stress and cell membrane integrity [13,14,15]. The literature reports cytotoxicity in HepG2 cells when exposed to graphene oxide (GO) and reduced graphene oxide (rGO), with an observed EC50 value of 50 µg/mL, which falls within the concentration range used in our study. Notably, graphene has been shown to increase the production of cytoplasmic reactive oxygen species (ROS) in human macrophages. However, contrasting responses have been observed between human and murine macrophages following graphene treatment [16,17]. 

Graphene nanoplatelets, with their varying lateral sizes, have increasingly been utilized to promote the growth and differentiation of mesenchymal stem cells (MSCs). With the expanding applications of MSCs and ongoing research into pristine graphene, the cytotoxicity of these materials has become a significant area of investigation. For instance, graphene oxide (GO) has demonstrated concentration-dependent cytotoxicity toward human MSCs (hMSCs). Specifically, GO does not exhibit cytotoxic effects at a concentration of 0.1 mg/mL. However, concentrations exceeding 0.1 mg/mL have been reported to induce cytotoxicity in hMSCs [18,19]. Increased oxidative stress and cell membrane damage have been linked to various factors, including the concentration, shape, and lateral size of graphene-based nanomaterials and the duration of exposure to mesenchymal stem cells (MSCs). For example, small reduced graphene oxide (rGO) nanoplatelets, even at a low concentration of 1.0 mg/mL, were found to cause significant damage to human MSCs (hMSCs) after a short exposure period. In contrast, larger graphene sheets exhibited cytotoxicity that was dependent on their shape and was only observed at high concentrations (100 mg/mL) and after extended exposure times [20].

The biocompatibility of graphene nanoplatelet biomaterials is essential for evaluating their potential impact on macrophage phagocytic activity. A significant finding from our study is related to BV–2 microglia. These lineages, characterized by their inherent phagocytic activity, exhibit pronounced interactions with exogenous particles, including graphene nanoplatelets, which can influence their functional responses. This observation aligns with recent research examining the in vitro interactions between blended graphene nanosheets, approximately 8 to 9 nm in size, and mesenchymal stem cells (MSCs) derived from mouse bone marrow [21]. Interesting data in these works indicated that the UGZ–1004 graphene nanoplatelets were efficiently internalized by MSCs, accumulating within the cytoplasm. Importantly for the cell fate, besides not exhibiting cytotoxicity, these nanomaterials also did not affect the expression of cellular surface markers in MSCs. Despite this, research into the mechanisms of graphene-based nanomaterial uptake by cells remains limited [22]. The primary pathway identified for the internalization of graphene, with lateral dimensions ranging from 100 nm to 1 µm, is predominantly micropinocytosis. Additionally, the internalization of graphene via macropinocytosis was consistent with findings from previous studies. Specifically, Linares et al. [23] and Chen et al. [20] reported that macropinocytosis is a significant mechanism for the uptake of graphene oxide with a lateral dimension of 100 nm in human liver cancer cells and human bone cancer cells, respectively. 

It has been observed that graphene nanosheets with dimensions in the range of hundreds of micrometers can induce concentration-dependent cytotoxicity. This effect is attributed to their hydrophobic interactions with the mammalian cell plasma membrane [20,24]. It is essential to monitor the accumulation of graphene nanosheets on the cellular plasma membrane, as this accumulation can disrupt intracellular homeostasis, including the redox balance, potentially leading to programmed cell death. After exposure to graphene, MSCs exhibited upregulation of genes such as CDK1, CDK2, KPNA2, CDC6, MDM2, and MRE11A. Notably, CDK1, CDK2, and MRE11A are key regulators of cell proliferation. CDK1 and CDK2, members of the cyclin-dependent kinase (Cdk) family, play crucial roles in regulating the transition between the G2 phase and mitosis. Specifically, CDK1 is closely involved in several critical cellular events essential for cell survival [25]. Numerous strategies have been employed to modulate cellular plasticity profiles. For instance, Marinowic et al. explored the effects of co-culturing skin fibroblasts with a mononuclear fraction of umbilical cord blood, directly and indirectly. This exposure resulted in a reduction in the expression of deleterious cell cycle control genes in the fibroblasts. This reduction was attributed to an enhancement in cellular plasticity induced by the co-culturing process. These findings are particularly significant given the potential application of MSCs in treating various human pathologies [26].

Low doses of radiation (75 mGy for 24 h) have been shown to modulate the proliferative capacity of MSCs by regulating CDK1. This radiation exposure stimulated proliferation and increased the proportion of MSCs in the S-phase of the cell cycle [27]. Similarly, controlled exposure to varying concentrations of zinc has been found to enhance the expression of CDK genes. These findings are associated with improved adhesion, migration, and self-renewal capabilities of MSCs [28].

The possible increase in cell proliferation and self-renewal observed in MSCs exposed to graphene may be associated with the elevated expression of the MDM2 gene. MDM2 encodes an oncoprotein that inhibits p53-mediated transcriptional activation [29]. Thus, after exposure to graphene, MSCs encounter direct and indirect proliferative stimuli by inhibiting cell cycle suppressors.

This increase in MDM2 gene expression relationship with cell proliferation is accompanied by elevated expression of the CDC6 and MRE11A genes. This is a significant finding because both genes play crucial roles in maintaining DNA integrity. CDC6 regulates the initiation of DNA replication, while MRE11A is essential for effective DNA repair and ensuring that replication is completed before mitosis begins [30,31].

The positive fold change observed in genes associated with cell proliferation, as well as those involved in checkpoint regulation and DNA repair, indicates that the interaction of mesenchymal cells with graphene nanoplatelets can boost the proliferative capacity of these cells. This enhancement seems to maintain essential mechanisms that protect against DNA damage during the transition from the S phase to the G1 phase and can affect the safety of cell expansion. The clinical application of MSCs involves extensive expansion and maintenance of the undifferentiated MSCs’ potential. Translational MSC research seeks an alternative to double the short stem cell life span, exhibit senescence in long-term culture in vitro, and lose the differentiation potentials with increasing time in culture [32,33]. 

Some genes exhibited a negative fold change after exposure to graphene, including CCNG2, CCND1, TFDP1, and CDC34. Notably, CCND1 and TFDP1 are genes whose overexpression is linked to the development of various types of tumors [34,35]. MSCs exposed to graphene exhibited downregulation of certain genes, which is a noteworthy finding in the context of tumor inhibition. Specifically, increased expression of TFDP1 has been linked to the enlargement of hepatocellular carcinoma, while its downregulation has been shown to inhibit the growth of Hep3B cells. This suggests that TFDP1 overexpression may contribute to tumor cell progression and growth [35].

BV–2 cells are crucial in regulating neural activity, synaptic plasticity, and neuroinflammation [36]. The gene expression after exposure of the UGZ–1004 nanoplatelets showed an 11.84-fold increase in the expression of the CASP3 gene, which plays a crucial role in neuronal apoptosis. Caspase–3 is a pivotal mediator in the apoptosis process, activated through both extrinsic (death ligands) and intrinsic (mitochondrial) pathways. Hyperactivation of caspase–3 and the resulting increase in cell death are linked to the development of neurodegenerative diseases, such as Huntington’s disease (HD) and Alzheimer’s disease (AD). Although exposure of BV2 cells to graphene did not induce cytotoxicity, it did enhance their apoptotic potential, as evidenced by the significant upregulation of CASP3 gene expression.

The CCNH and MCM4 genes also exhibited significant positive fold changes following graphene exposure in BV2 cells. These genes play critical roles in transcriptional control and helicase functionality, essential for DNA recognition and replication initiation. Conversely, the negative cell cycle regulator gene CCNG2 showed a notable 9.46-fold increase in expression. Elevated levels of CCNG2 can inhibit cell proliferation by inducing cell cycle arrest between the G0 and G1 phases and may even promote apoptosis [37].

After exposure to graphene, BV–2 cells demonstrated significant alterations in gene expression profiles. These changes suggest a biological response where cells initially prepare for division, but this process is subsequently inhibited due to cell cycle arrest, leading to apoptosis. These findings highlight the potential safety of MSCs concerning tumorigenesis, suggesting that graphene exposure does not compromise their potential for clinical applications. Due to their ability to migrate to injured sites in response to environmental signals and to promote tissue regeneration through the release of anti-inflammatory and growth factors, MSCs are obtained through a minimally invasive method. This patient-specific approach positions MSCs as a promising tool in regenerative medicine [38].

The microenvironment can significantly impact the metabolism and phenotype of various cell types. Therefore, understanding the interaction between the chemical composition of graphene and fundamental cellular mechanisms is crucial. This understanding should be continually evaluated, given this versatile material’s diverse production methods and applications. This study emphasizes the non-cytotoxic nature of graphene. Also, it provides insights into genes related to the cell cycle, which is the first step in determining cell fate. It demonstrates that exposure to graphene nanoplatelets can modulate genes that enhance mammalian cell proliferation and self-renewal in MSCs. Both main points set a precedent for further investigating data about the fate of MSCs as a potential approach to managing and controlling MSCs’ behavior as a cell therapy product for medical use.

The positive fold changes in genes associated with cell proliferation stability, including checkpoint and DNA repair genes, suggest that graphene nanoplatelets can safely enhance the proliferative capacity of MSCs. Conversely, the observed negative fold changes in genes related to tumor development imply that interactions between graphene and MSCs do not lead to overexpression of these tumorigenic genes. This indicates effective control of cell fate and reduces the potential risk of tumorigenesis often linked with undifferentiated cells. However, it is important to note that this study is preliminary and limited to gene expression analysis; thus, the phenotypic changes related to the proliferation and self-renewal of MSCs have not yet been conclusively demonstrated.

The literature indicates that graphene is not a homogeneous material; instead, each type of graphene can elicit a highly specific molecular response. Consequently, the impact of graphene nanoplatelets, such as UGZ–1004, on gene expression is influenced by exposure time, dose, and cell type. It is crucial to determine whether the identified mechanisms are transient or if they might have long-term effects on cell fate and health. Future studies should utilize relevant models, such as neurodegenerative disease models or joint trauma/musculoskeletal models, to thoroughly explore the effects of long-term exposure to graphene nanoplatelets. Additionally, future research should extend beyond single-exposure scenarios to encompass studies involving repeated and prolonged exposure in both in vivo and preclinical models. The novel insights into the intracellular mechanisms of graphene nanoplatelets in mammalian cells and MSCs are crucial for advancing research and ensuring the safe application of these promising biomaterials.

## 4. Materials and Methods

### 4.1. Graphene Nanoplatelets

The graphene nanoplatelets used in this study were produced via the liquid-phase exfoliation method. This technique is known for its effectiveness in generating colloidal graphene suspensions from graphite and is considered the most promising approach for large-scale production [39]. The method involves separating graphite layers with a hexagonal structure using a liquid chemical. Graphite consists of multiple graphene layers stacked together, held by weak π-π interactions between the layers. The liquid medium aids in disrupting these interactions, enabling the separation of the graphene layers. An external force is required to allow the molecules of the liquid medium to penetrate between the layers and break the π-π bonds [40].

The graphene nanoplatelets were characterized using atomic force microscopy (AFM) and scanning electron microscopy (SEM) to determine their thickness and lateral size, respectively. AFM measurements were conducted with a Shimadzu® SPM–9700HT microscope (Nishinokyo Kuwabara-cho, Nakagyoku, Kyoto, Japan) following the ABNT ISO/TS 21356–1 standard. SEM analyses were performed using a Tescan MIRA3 (Libusina Trida, Brno, Czech Republic), also in accordance with ABNT ISO/TS 21356–1 standard procedures. Raman spectroscopy tests were performed in accordance with the ABNT ISO/TS 21356–1 standard using a Horiba LabRAM HR Evolution spectrometer equipped with a 633 nm wavelength laser. To characterize the chemical composition of the graphene nanoplatelets, carbon purity was analyzed following the ISO 11,308 standard (Nanotechnologies: Characterization of single-wall carbon nanotubes using thermogravimetric analysis). This analysis was conducted using a Netzsch STA 449 F3 Jupiter^®^ thermogravimetric analyzer (Netzsch, Selb, Germany), under an airflow of 100 mL/min, from room temperature to 1000 °C, with a heating rate of 5 °C/min. An alumina crucible was used for the analysis. According to the ISO 11,308 standard, the purity content is correlated with the residual mass remaining in the crucible at the end of the analysis. 

Energy-dispersive X-ray spectroscopy (EDS) analysis was also performed to identify the main constituents of the sample. Initially, the graphene nanoplatelet sample was pelletized for semi-quantitative analysis. Approximately 0.04 g of graphene nanoplatelets was weighed and then placed into a mold to form the pellet. The mold was adjusted, and the assembly was subjected to a press, which applied a pressure of four tons for approximately 30 min. After pressing, the pellet was removed and stored in a desiccator for subsequent analysis. The use of pellets, as opposed to powdered samples, was intended to minimize interference from materials such as carbon tape, which is used to secure the sample to the stub and could otherwise affect the semi-quantitative results.

### 4.2. General Cell Culture Protocols

VERO (African green monkey kidney epithelial cells), 3T3 (murine fibroblasts), and BV–2 (murine microglia) cell lines were obtained from the American Type Culture Collection (ATCC, Rockville, MD, USA). These cells were cultured in Dulbecco’s Modified Eagle Medium (DMEM) supplemented with 10% fetal bovine serum (FBS), 100 U/mL penicillin, and 100 µg/mL streptomycin at 37 °C in a 5% CO_2_ atmosphere. Mesenchymal stem cells (MSCs) were sourced from equine bone marrow progenitor cells and cultured in tissue culture flasks with a growth area of 75 cm^2^, using DMEM supplemented with 10% FBS and penicillin/streptomycin (50 U/mL and 50 µg/mL, respectively; Gibco-Invitrogen, Carlsbad, CA, USA) until reaching 80% confluence. These cells were then passaged to passage 5. The culture plates were maintained at 37 °C in a humidified 5% CO_2_ atmosphere for 48 h. For medium exchange, the plates were washed with PBS to remove non-adherent cells, and the medium was then replaced.

All cell lines were pre-seeded at a density of 3–5 × 10^3^ cells per well in 96-well plates or 15–20 × 10^3^ cells per well in 24-well plates, depending on the experimental protocol. Prior to experimentation, the graphene nanoplatelet biomaterial, referred to as UGZ–1004, was subjected to UV light for 30 min on each side in a Class II safety cabinet as part of the decontamination protocol. VERO, 3T3, BV–2, and MSC cells were exposed to UGZ–1004 using both direct contact and elution methods at a concentration of 6 cm^2^/mL, in accordance with ISO 10993–1 (2023) [41] standards for the biological evaluation of sterile medical devices in contact with the human body. The MTT and DAPI assays were conducted in triplicate and repeated three times, as recommended by ISO 10993–1 (2023).

### 4.3. MSCs Characterization by Flow Cytometry

To characterize mesenchymal stem cells (MSCs), surface markers recommended as minimum characterization criteria by the International Society for Cellular Therapy (ISCT) were utilized. MSCs at passage 5 were characterized both before and after exposure to graphene. Surface antigens were detected by incubating MSCs with antibodies against CD44, CD90, CD105, CD45, and CD34. Following labeling, the cells were analyzed using flow cytometry with a FACSCanto cytometer (BD—Becton, Dickinson and Company, Franklin Lakes, NJ, USA).

### 4.4. Cytotoxic Analysis by MTT 

Cell viability was assessed using the MTT assay to measure the production of formazan [3-(4,5-dimethylthiazol–2-yl)–2,5-diphenyl–2H-tetrazolium] bromide (Sigma Aldrich, St. Louis, MO, USA). Following exposure to UGZ–1004 for 24, 48, and 72 h, 3T3, VERO, BV–2, and MSC cells were incubated with a 0.5 mg/mL MTT solution diluted in PBS (pH 7.4) for three hours at 37 °C, protected from light. After incubation, the cells were treated with 300 µL/well of dimethyl sulfoxide (DMSO) (Sigma-Aldrich, St. Louis, MO, USA) to solubilize the formazan crystals. The absorbance was then measured at 570 nm using a Bio-Rad spectrophotometer (Bio-Rad Laboratories, Hercules, CA, USA). Absorbance values were converted into percentages using the following formula:(1)$$\mathrm{Percentage}\;\mathrm{Viability}=\left(\frac{\mathrm{Mean}\;\mathrm{Absorbance}\;\mathrm{of}\;\mathrm{Treatments}\times 100}{\mathrm{Mean}\;\mathrm{Absorbance}\;\mathrm{of}\;\mathrm{Control}}\right)$$

The data were analyzed using a one-way analysis of variance (ANOVA) followed by Bonferroni’s post hoc test with GraphPad Prism 8 Software (San Diego, CA, USA). The results are presented as the mean ± standard deviation. A *p*-value of less than 0.05 was considered indicative of statistical significance compared to the control group.

### 4.5. Nuclear Analyses by DAPI

The 4′,6-diamidino–2-phenylindole (DAPI) staining method provides detailed insights into nuclear morphology, including area, roundness, and solidity, which are related to various cell survival mechanisms. DAPI staining was used to assess nuclear morphology and determine cell proliferation rates in BV–2 cell lines. Following exposure to UGZ–1004 for 24, 48, and 72 h, cells were washed three times with 1% PBS and fixed with 4% formaldehyde at room temperature for 15 min. The fixed cells were then washed again with 1% PBS, permeabilized with 0.1% Triton X–100 in 1% PBS, and stained with a 300 nM DAPI solution (Santa Cruz, CA, USA) at room temperature for 10 min.

Nuclear morphology was examined under a fluorescent microscope (Carl Zeiss MicroImaging GmbH, Oberkochen, Germany). DAPI staining highlighted the nuclear morphology, enabling the quantification of nuclear roundness and solidity using ImageJ Software version 1.54j, with ten fields analyzed per sample. Data from control cells (not exposed to UGZ–1004) were used to establish baseline parameters. The experiments were analyzed using a one-way analysis of variance (ANOVA) followed by Bonferroni’s post hoc test with GraphPad Prism 8 Software. The results are reported as the mean ± standard deviation. A *p*-value of less than 0.05 was considered statistically significant.

### 4.6. Live/Dead Cell Assay

The LIVE/DEAD Cell Viability Assay kit (Thermo Fischer Scientific, Waltham, MA, USA) was used to identify and quantify dead cells in mammalian cell types exposed to UGZ–1004 for 48 h. BV–2 cells were labeled according to the manufacturer’s instructions. In this assay, cells were incubated with Calcein-AM, which stains live cells green, and propidium iodide (PI), which stains dead cells red. Following incubation, the cells were returned to the incubator for an additional 30 min to allow for complete permeation of Calcein-AM and PI. Images were then acquired from four regions of gray matter in each group, with three images taken per region in triplicate. Image and data analyses were performed using an AXIOVERT II inverted fluorescence microscope (Carl Zeiss MicroImaging GmbH, Oberkochen, Germany) and Zen Blue software 2.3. 

### 4.7. Real-Time qRT-PCR

RNA was extracted from MSC and BV–2 cells after 48 h of exposure to UGZ–1004 using the SV-Total RNA Isolation System kit (Promega, Madison, WI, USA) according to the manufacturer’s instructions. The cells were lysed with a lysis buffer, heated to 70 °C for 3 min, and then centrifuged at 12,000× *g* for 10 min at 4 °C. The supernatant was collected, and 200 µL of 95% ethyl alcohol was added. The mixture was transferred to a spin column and centrifuged at 12,000× *g* for 1 min. RNA was eluted from the columns with 100 µL of RNase-free water by centrifugation at 12,000× *g* for 2 min.

Complementary DNA (cDNA) synthesis was performed using the GoScript Reverse Transcriptase kit (Promega, Madison, WI, USA) following the manufacturer’s instructions. The resulting cDNA was used for RT^2^ Profiler™ PCR Array Human Cell Cycle (SuperArray Bioscience Corporation, Frederick, MD, USA), which includes 92 primers targeting gene regions related to the cell cycle, as well as 4 endogenous gene controls.

### 4.8. Population Doubling Level (PDL)

BV–2 cells were exposed to UGZ–1004 graphene nanoplatelets and cultured until reaching approximately 70% confluence. Cells were plated and cultured at intervals of 3, 7, 10, 21, and 28 days, with 70% confluence serving as the criterion for trypsinization. Half of the cell density was transferred to new wells with fresh medium at each trypsinization.

The Population Doubling Level (PDL) was calculated using ImageJ Software. Since graphene nanoplatelets appear gray under light microscopy, they facilitated the analysis of UGZ–1004 retention and depletion over the cultivation period. Digitized RGB (24-bit) images were captured and analyzed using a custom macro to quantify positive areas based on pixel color. This macro enabled the recognition and tracking of graphene nanoplatelets until their maximum depletion from the culture. The macro was applied to all images obtained during the experimental period, and results regarding optical density were recorded.

## Figures and Tables

**Figure 1 ijms-25-09817-f001:**
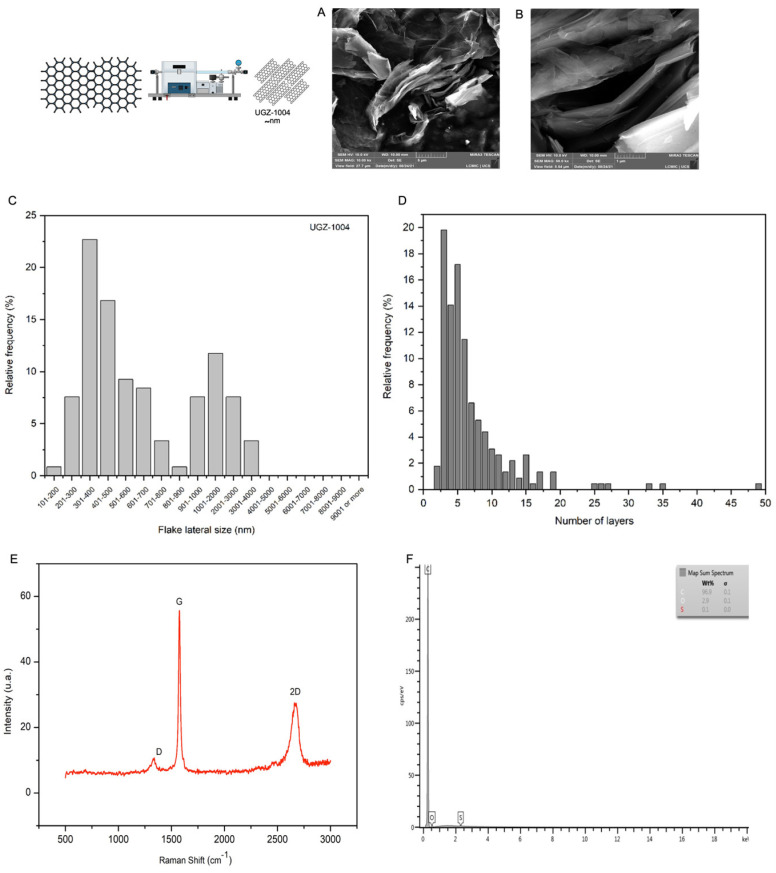
Graphene nanoplatelet characterization. (**A**,**B**) Scanning electron micrograph of UGZ–1004 sample obtained at 10,000× magnification and image obtained at 50,000× magnification. (**C**) Relative frequency histogram of the number of layers of the UGZ–1004 sample. (**D**) Relative frequency histogram of lateral size of the UGZ–1004 sample. (**E**) Raman spectrum of the UGZ–1004 sample. (**F**) EDS spectrum of the UGZ–1004 sample.

**Figure 2 ijms-25-09817-f002:**
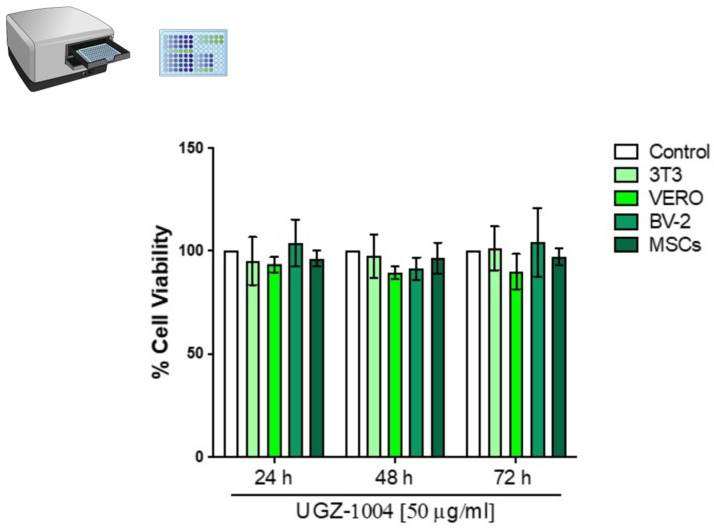
Cell viability of 3T3, VERO, BV–2, and MSCs after exposure to UGZ–1004 for 24, 48, and 72 h. The results are expressed as a percentage relative to the control, which was set at 100%. Data represent the mean of three independent experiments (*n* = 3).

**Figure 3 ijms-25-09817-f003:**
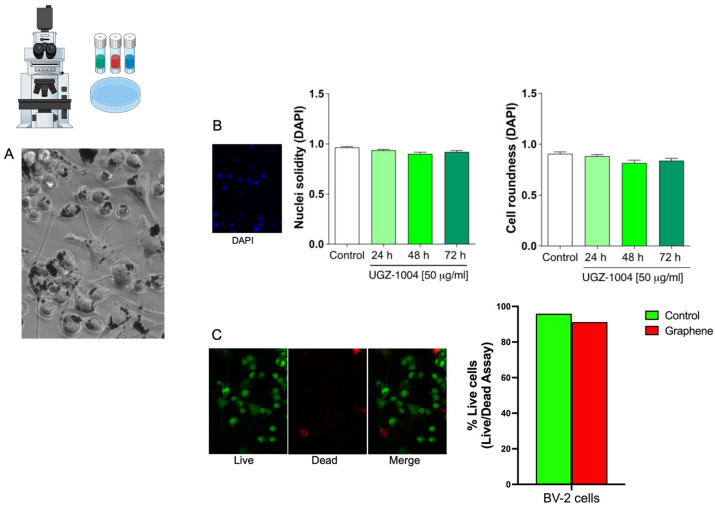
Intracellular interaction and viability. (**A**) BV2 microglia cells under cultivation plus UGZ1004 formulation. Internalized graphene nanoplatelets and maintenance of the usual proliferative profile and morphology (40× magnification). (**B**) Representative histogram of nuclei solidity and cell roundness marker by DAPI. (**C**) Live/dead assay showing the cells stained green (indicative of cell viability) and red stains (indicate dead cells).

**Figure 4 ijms-25-09817-f004:**
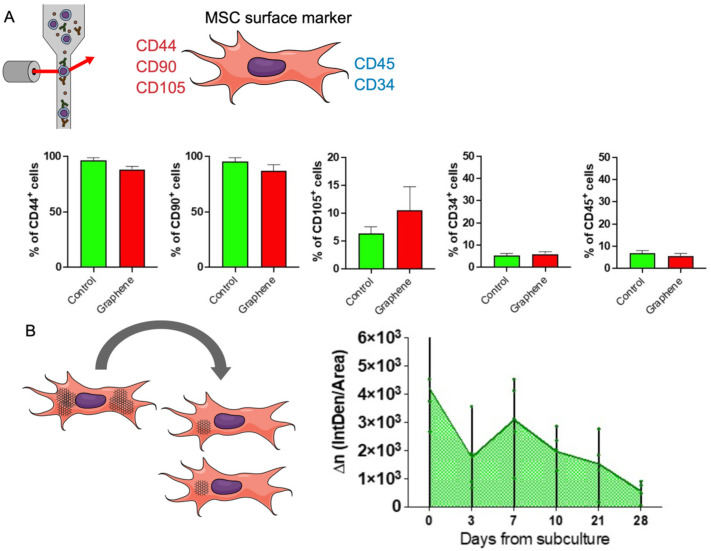
Representation of MSC immunophenotyping and graphene behavior in subcultures. (**A**) Surface marker profile in MSCs: MSCs were analyzed for their surface marker expression before and after exposure to UGZ–1004. Flow cytometry results show that MSCs maintained their characteristic surface marker profile, including anti-CD44, anti-CD90, anti-CD105, anti-CD45, and anti-CD34, demonstrating that mesenchymal stem cell properties were preserved despite graphene nanoplatelet exposure. (**B**) Population Doubling Level (PDL) in BV–2 Cells: The UGZ–1004 incorporated into the cytoplasm of cells decreases with each cell division (illustration). BV–2 microglia cells were cultured with UGZ–1004 at 3, 7, 10, 21, and 28 days. The Population Doubling Level (PDL) was measured to assess cell growth and proliferation over time. The results indicate that BV–2 cells were able to proliferate and maintain their growth profile despite the presence of graphene nanoplatelets.

**Figure 5 ijms-25-09817-f005:**
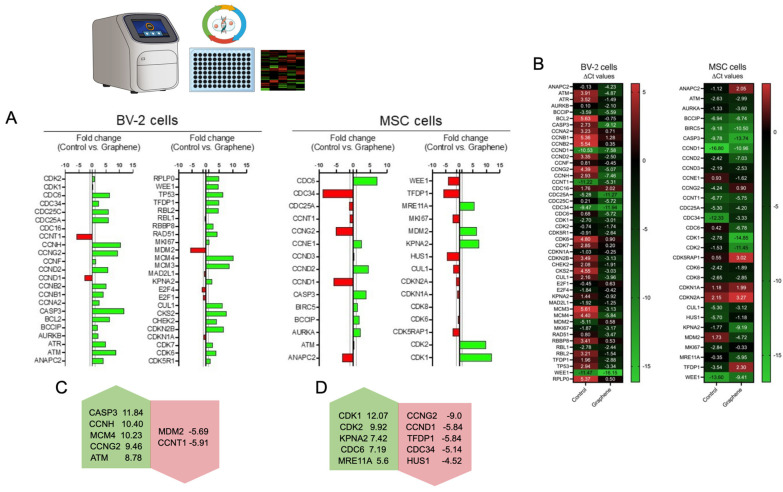
Molecular analysis of BV2 and MSC. Assessment of differences in gene expression profiles following exposure to graphene. (**A**) Relative expression analysis using the fold change technique. (**B**) Heat maps showing the Delta Ct values for each gene analyzed. The fold change values for the genes with the highest and lowest differences are highlighted for BV–2 cells (**C**) and MSC cells (**D**).

## Data Availability

The original contributions presented in the study are included in the article, further inquiries can be directed to the corresponding author.

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
