# Peer review of "Non-Cytotoxic Graphene Nanoplatelets Upregulate Cell Proliferation and Self-Renewal Genes of Mesenchymal Stem Cells"

_ijms, 2024, doi:10.3390/ijms25189817_

Round 1

Reviewer 1 Report

Comments and Suggestions for Authors

Nicoletti et al. present a manuscript that attempts to address a broad topic with major implications. The biomaterial proposed by the authors has extensive research supporting its potential use. For this reason, the authors should have made a greater effort to justify the novelty and implications of this study. However, the authors limit themselves to performing a simple proof of concept, in a model with very few plausible conclusions. This study is too simplistic for the categorical conclusions that the authors claim.

When evaluating the manuscript, the reader thinks only from the summary that we have a study of a real preclinical model. However, this is not true. The reality is that very limited tests are used and with a very small sample size. The quality of the findings is discrete and in some cases too simple. The discussion is non-existent, with no proposals. The conclusions are not supported by the results shown.

Comments on the Quality of English Language

 Extensive editing of English language required.

Author Response

Reviewer 1

Nicoletti et al. present a manuscript that attempts to address a broad topic with major implications. The biomaterial proposed by the authors has extensive research supporting its potential use. For this reason, the authors should have made a greater effort to justify the novelty and implications of this study. However, the authors limit themselves to performing a simple proof of concept, in a model with very few plausible conclusions. This study is too simplistic for the categorical conclusions that the authors claim.

Answer: Thank you for your contribution and comments. Our article is subdivided into a stage of non-maleficence analysis and compatibility of the UGZ-1004 graphene presentation, as well as an exploration of the gene expression profile in mesenchymal stem cells after exposure. The super array methodology presents a targeted screening result (in this case, cell cycle-related genes) and allows for the generation of targets for further studies. We understand that protein quantification and cellular function analyses would greatly contribute to consolidating the results; however, we believe we have thoroughly explored the results at the molecular level.

When evaluating the manuscript, the reader thinks only from the summary that we have a study of a real preclinical model. However, this is not true. The reality is that very limited tests are used and with a very small sample size. The quality of the findings is discrete and in some cases too simple. The discussion is non-existent, with no proposals. The conclusions are not supported by the results shown.

Answer: As described in the previous response, our objective was to describe the cytotoxicity and action of this graphene presentation in mesenchymal cells. We made some changes to the discussion and conclusion (highlighted) to improve the text and address this issue.

Reviewer 2 Report

Comments and Suggestions for Authors

The authors have reported on the cell proliferation upregulation and regeneration of MSCs using graphene nanoplates. Although this manuscript could be of interest to the readers of this journal, there are a few issues that need to be addressed before suggesting for publication.

The overall English of this manuscript needs to be improved.

The introduction of this manuscript should be strengthened by adding other examples of using graphene for cell proliferation in different stem cell types such as 10.1088/1748-605X/ab8d12

The findings of this research should be discussed briefly in the introduction.

What is the justification of using 50 ug/ml of UGZ-1004 in the MTT assay and not any other concentration?

Could the authors provide some characterization results of the UGZ-1004 such as Raman, EDX or SEM images?

Comments on the Quality of English Language

The overall English of this manuscript needs to be improved.

Author Response

Reviewer 2:

The authors have reported on the cell proliferation upregulation and regeneration of MSCs using graphene nanoplates. Although this manuscript could be of interest to the readers of this journal, there are a few issues that need to be addressed before suggesting for publication.

Answer: We would like to thank you very much for the encouraging comments on our study. We really hope that revised version of our manuscript has been satisfactorily improved to result in the acceptance of our manuscript in this occasion.

The overall English of this manuscript needs to be improved.

Answer: The English language was verified by a native speaker in the revised version of our manuscript.

The introduction of this manuscript should be strengthened by adding other examples of using graphene for cell proliferation in different stem cell types such as 10.1088/1748-605X/ab8d12.

Answer: As required by this reviewer, we read the suggested article and we consider it more appropriate to insert this reference in the discussion section (page 11, line 379) of the manuscript.

The findings of this research should be discussed briefly in the introduction.

Answer: Thank you for the suggestion, we have described the mainly findings of this research in the introduction of the new version of our manuscript (page 2, lines 78-82), as recommended by the reviewer.

What is the justification of using 50 ug/ml of UGZ-1004 in the MTT assay and not any other concentration?

Answer: Previous publications have already tested and supported the time-dependent toxic and non-toxic concentrations from graphene nanoplatelets and graphene-based materials in different cell types (doi.org/10.1016/j.biomaterials.2014.02.054; doi: 10.3390/nano12121978). Based on these researches findings we selected the most common non-toxic concentration used to carry our assays settings.

Could the authors provide some characterization results of the UGZ-1004 such as Raman, EDX or SEM images?

Answer: Thank you very much for your relevant suggestion. Please, note that we have provided novel data analysis which include Raman and EDX characterization and SEM images. Please, note that we have added this information to the revised version of our manuscript in the methods and results sections (page 3, lines 99-101; page 3, lines 109-119; pages 6-7, lines 246-267).

Round 2

Reviewer 1 Report

Comments and Suggestions for Authors

The authors present the revised version of the manuscript ijms-3150377. The authors have realized minimal changes and aesthetics. The authors have not realized any significant changes, the images are of low quality and the conclusions are transversal and not specified. Authors have not improved grammatical aspects. This manuscript does not contain the minimum calibration levels for this Journal.

Comments on the Quality of English Language

 Extensive editing of English language required.

Author Response

Coment: The authors present the revised version of the manuscript ijms-3150377. The authors have realized minimal changes and aesthetics. The authors have not realized any significant changes, the images are of low quality and the conclusions are transversal and not specified. Authors have not improved grammatical aspects. This manuscript does not contain the minimum calibration levels for this Journal.

Answer: Thank you for your review. Sincerely, in the first round of revision, the comments and questions you provided were somewhat indirect. We were unable to pinpoint specific comments to give a convincing and satisfactory response. We have thoroughly revised the entire manuscript again to improve it and address your requests. The previous revisions are highlighted in yellow, and the new revisions are in green.

All images in the manuscript were originally inserted at 300 DPI. However, we have now revised the images and saved them at 1200 DPI to enhance their quality.

The English language editing was done by Professor Edy Lorraine Hoffman (certificate attached).

Let me know if you need any further adjustments!

Round 3

Reviewer 1 Report

Comments and Suggestions for Authors

The revised version of the manuscript is not satisfactory. This manuscript still does not meet the requirements of this journal.

Comments on the Quality of English Language

Extensive editing of English language required.

Author Response

Reviewer 2

The revised version of the manuscript is not satisfactory. This manuscript still does not meet the requirements of this journal.

Answer: Honestly, we apologize for the unsuccessful attempts to respond to the reviewer. However, the review did not specifically point out any shortcomings in the article, such as a sentence, a page, or a line, as is usually the case in peer reviews. We have tried to respond to Reviewer 2 by interpreting their comments, as from the first review, they were presented in a general manner. At this time, we have addressed the editor’s questions point by point, as usual.

If Reviewer 2 would like to provide a point-by-point review, we will certainly make every effort to respond thoroughly.
